# A dual-target herbicidal inhibitor of lysine biosynthesis

Emily RR Mackie[1,2], Andrew S Barrow[1†], Rebecca M Christoff[3†],
Belinda M Abbott[3], Anthony R Gendall[4,5], Tatiana P Soares da Costa[1,2]*

[1]Department of Biochemistry and Genetics, La Trobe Institute for Molecular Science,
La Trobe University, Bundoora, Australia; [2]School of Agriculture, Food and Wine,
Waite Research Institute, The University of Adelaide, Waite Campus, Glen Osmond,
Australia; [3]Department of Chemistry and Physics, La Trobe Institute for Molecular
Science, La Trobe University, Bundoora, Australia; [4]Australian Research Council
Industrial Transformation Research Hub for Medicinal Agriculture, AgriBio, La Trobe
University, Bundoora, Australia; [5]Department of Animal, Plant and Soil Sciences, La
Trobe University, Bundoora, Australia

**Abstract** Herbicides with novel modes of action are urgently needed to safeguard global agricultural industries against the damaging effects of herbicide-resistant weeds. We recently developed the first herbicidal inhibitors of lysine biosynthesis, which provided proof-of-concept for a promising novel herbicide target. In this study, we expanded upon our understanding of the mode of action of herbicidal lysine biosynthesis inhibitors. We previously postulated that these inhibitors may act as proherbicides. Here, we show this is not the case. We report an additional mode of action of these inhibitors, through their inhibition of a second lysine biosynthesis enzyme, and investigate the molecular determinants of inhibition. Furthermore, we extend our herbicidal activity analyses to include a weed species of global significance.

*For correspondence:
tatiana.soaresdacosta@adelaide.
edu.au

†These authors contributed
equally to this work

Reviewing Editor: Todd Gaines,
Colorado State University, United
States

## Editor's evaluation

This paper presents the highly interesting and novel finding that the investigated compound inhibits two targets in lysine synthesis. Further, the first enzyme has allosteric inhibition, while the second enzyme is a competitive inhibitor. The authors do a good job explaining why this is high interest for herbicide resistance management in a new compound. The authors demonstrated that the compound is not a pro-herbicide, and propose that the higher in vivo activity relative to in vitro activity is due to the simultaneous in vivo inhibition of two separate steps in lysine synthesis.

## Introduction

Effective herbicides are critical for sustainable agriculture. However, our current options are dwindling as the prevalence of herbicide-resistant weeds continues to rise (*Hall et al., 2020*). Weeds have now evolved resistance to 21 out of the 31 herbicide modes of action, yet there has been a lack of herbicides with new modes of action brought to market over the last 30 years (*Duke and Dayan, 2022*; *Heap, 2022*).

Despite the success of targeting amino acid biosynthesis enzymes for the development of herbicides (e.g. glyphosate and chlorsulfuron), the inhibition of plant lysine biosynthesis has never been explored commercially. Our group was the first to discover inhibitors of lysine biosynthesis with herbicidal activity (*Soares da Costa et al., 2021*). We showed that the most potent of these inhibitors, (*Z*)-2-(5-(4-methoxybenzylidene)-2,4-dioxothiazolidin-3-yl)acetic acid (MBDTA-2) (*Figure 1A*) targets

**Figure 1.** Structure and mode of binding of (*Z*)-2-(5-(4-methoxybenzylidene)-2,4-dioxothiazolidin-3-yl)acetic acid (MBDTA-2). (**A**) Chemical structure of MBDTA-2. (**B**) The AtDHDPS1 quaternary structure with MBDTA-2 (green sticks) bound within a novel allosteric pocket (PDB ID: 7MDS) (*Soares da Costa et al., 2021*).

lysine production by inhibiting dihydrodipicolinate synthase (DHDPS), the enzyme that catalyses the first and rate-limiting step in the pathway (*Soares da Costa et al., 2018*). Interestingly, we found that the mode of DHDPS inhibition by MBDTA-2 was through binding at a novel allosteric site distinct from the allosteric lysine-binding site (*Figure 1B*), which enables regulation of the enzyme (*Hall et al., 2021*).

In the previous study, we revealed that the *in vitro* potency of MBDTA-2 against recombinant *Arabidopsis thaliana* (At) DHDPS enzymes was similar to the activity against agar-grown *A. thaliana* (*Soares da Costa et al., 2021*). Usually, herbicides inhibit their enzyme targets with greater potency than they inhibit *in vivo* plant growth. This is because the amount of herbicide reaching the target site is less than the amount applied. We proposed that this unusual similarity between *in vitro* and *in vivo* activity may be due to MBDTA-2 acting as a proherbicide, which is modified *in vivo* to a form that is more active at the target site. Proherbicides have been reported in the literature such as diclofop-methyl, which is demethylated via ester hydrolysis *in vivo* to produce the more active compound diclofop . Whilst MBDTA-2 could not undergo the same process as it does not contain an ester, we postulated that a similar proherbicidal effect may be observed through the *in vivo* demethylation of the aryl methyl ether.

The present study sought to extend our understanding of the mode of action of our previously developed DHDPS inhibitors (*Christoff et al., 2021*; *Soares da Costa et al., 2021*). Specifically, we used biochemical enzyme kinetic assays to demonstrate that MBDTA-2 does not act as a proherbicide, and that the apparent similarity between *in vitro* and *in vivo* potency may instead be explained by this series of compounds having a novel, dual-target mode of action through inhibition of the second lysine biosynthesis enzyme in the pathway, dihydrodipicolinate reductase (DHDPR). Static docking and biochemical assays revealed that in contrast to the allosteric binding of these inhibitors to DHDPS, active site binding is responsible for their inhibition of DHDPR. Additionally, we extended our previous *in vivo* activity studies on *A. thaliana* to include one of the most agriculturally problematic weeds in the world, rigid ryegrass (*Lolium rigidum*) (*Bajwa et al., 2021*; *Busi and Beckie, 2020*).

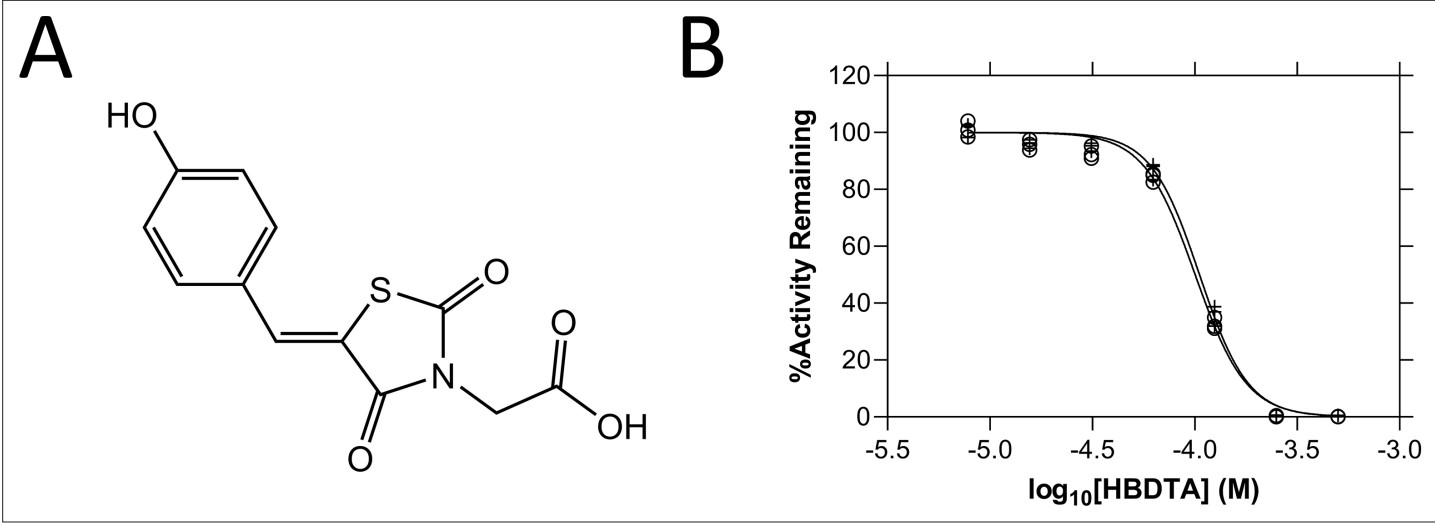

**Figure 2.** Structure and *in vitro* potency of (*Z*)-2-(5-(4-hydroxybenzylidene)-2,4-dioxothiazolidin-3-yl)acetic acid (HBDTA). (**A**) Chemical structure of HBDTA. (**B**) Dose–response curves of HBDTA against recombinant AtDHDPS1 (○) and AtDHDPS2 (+) enzymes. Initial enzyme rate was normalised against the vehicle control to determine % activity remaining. Data were fitted to a nonlinear regression model (solid line), resulting in $R^2$ values of 0.99.

The online version of this article includes the following source data for figure 2:

**Source data 1.** Source data for *Figure 2*.

## Results

### Inhibitory activity of a demethylated MBDTA analogue

Given that proherbicides are metabolised *in vivo* to produce compounds with greater potency at the target site, it can be assumed that the metabolised form will have greater activity than the proherbicidal form against the target *in vitro*. As such, to assess whether MBDTA-2 is a proherbicide that is demethylated *in vivo*, we measured the inhibitory activity of the demethylated analogue, (*Z*)-2-(5-(4-hydroxybenzylidene)-2,4-dioxothiazolidin-3-yl)acetic acid (HBDTA), against both recombinant *A. thaliana* DHDPS enzymes (*Figure 2*; *Christoff et al., 2021*). The dose–response curves yielded $IC_{50}$ values for AtDHDPS1 and AtDHDPS2 of 100 ± 0.95 and 105 ± 1.04 µM, respectively (*Figure 2*). These values are slightly greater than those we reported for MBDTA-2 ($IC_{50}$ (AtDHDPS1) = 63.3 ± 1.80 µM, $IC_{50}$ (AtDHDPS2) = 64.0 ± 1.00 µM) (*Soares da Costa et al., 2021*). These data suggest, conversely to our hypothesis, that the activity of MBDTA-2 at the target site is not influenced by the retention or loss of the methyl group *in vivo*.

### Dual-target activity of MBDTA-2

Given that the similarity in the *in vitro* and *in vivo* activity of MBDTA-2 could not be explained by enhanced target site activity of the demethylated compound, we sought to explore other mechanisms that may explain this observation.

We investigated whether additional modes of action beyond the inhibition of the DHDPS enzyme may account for the increased *in vivo* potency relative to the *in vitro* activity against the target enzyme. It is well established that proteins catalysing sequential reactions within a metabolic pathway often have conserved binding site features (*Hsu et al., 2013*; *Jensen, 1976*; *Jenwitheesuk et al., 2008*; *Zhang et al., 2009*). As such, we hypothesised that DHDPS inhibitors may also have activity against the enzyme following DHDPS in the plant lysine biosynthesis pathway, DHDPR. The activity of both recombinant *A. thaliana* DHDPR enzymes was measured whilst titrating MBDTA-2 to determine the $IC_{50}$ values of 6.92 ± 0.92 µM against AtDHDPR1 and 8.58 ± 1.19 µM against AtDHDPR2 (*Figure 3A*). Given that these results revealed a new target site for MBDTA-2, we assessed whether the compound could be proherbicidal with respect to DHDPR. To do so, we assessed the inhibitory activity of HBDTA against both AtDHDPR isoforms to determine the $IC_{50}$ values of 169 ± 0.92 and 155 ± 0.90 µM for AtDHDPR1 and AtDHDPR2, respectively (*Figure 3B*). HBDTA is therefore >18-fold less potent at the DHDPR target site relative to MBDTA-2, suggesting that the MBDTA-2 methyl group is beneficial for

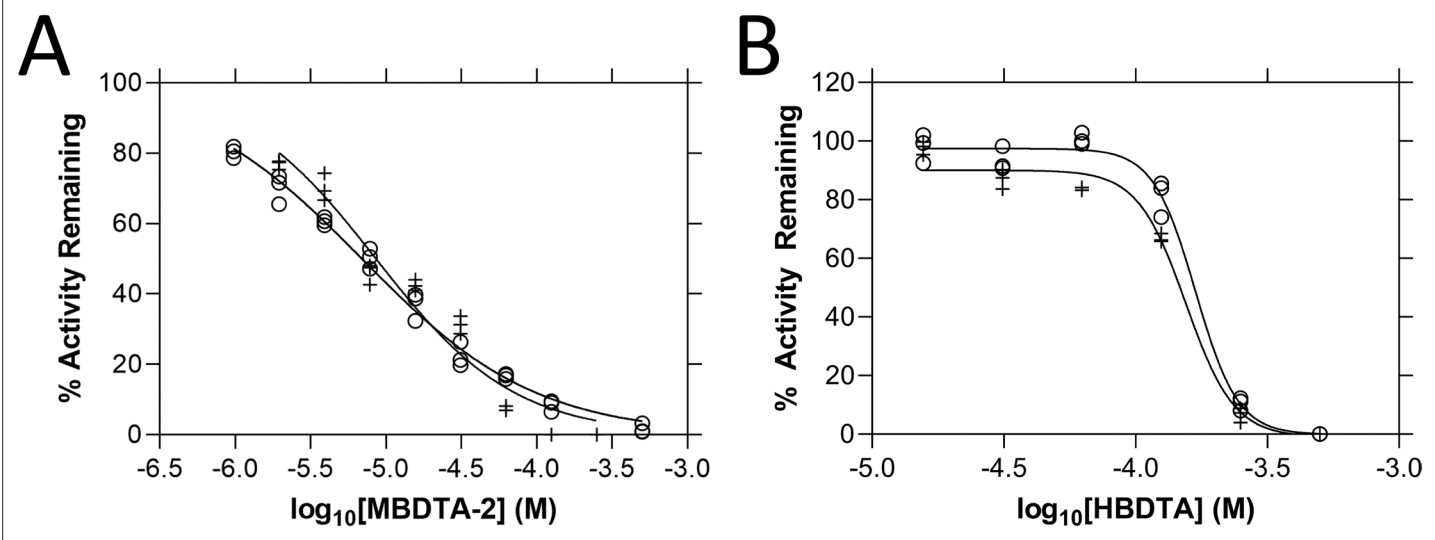

**Figure 3.** *In vitro* potency of (*Z*)-2-(5-(4-methoxybenzylidene)-2,4-dioxothiazolidin-3-yl)acetic acid (MBDTA-2) and (*Z*)-2-(5-(4-hydroxybenzylidene)-2,4-dioxothiazolidin-3-yl)acetic acid (HBDTA) against AtDHDPR. Dose–response curves of (**A**) MBDTA-2 and (**B**) HBDTA against recombinant AtDHDPR1 (○) and AtDHDPR2 (+) enzymes. Initial enzyme rate was normalised against the vehicle control to determine % activity remaining. Data were fitted to a nonlinear regression model (solid line), resulting in $R^2$ values of (**A**) 0.99 and 0.95 and (**B**) 0.99 and 0.99 for AtDHDPR1 and AtDHDPR2, respectively.

The online version of this article includes the following source data for figure 3:

**Source data 1.** Source data for *Figure 3*.

DHDPR inhibition. As such, MBDTA-2 is unlikely to be a proherbicide. Moreover, these results demonstrate that MBDTA-2 is a multi-targeted inhibitor of two consecutive enzymes in the lysine biosynthesis pathway, AtDHDPS and AtDHDPR. This compound represents the first example of a dual-target inhibitor of the lysine biosynthesis pathway.

## Mode of AtDHDPR inhibition by MBDTA-2

To investigate the molecular determinants of inhibition of AtDHDPR, we sought to co-crystallise the enzyme with MBDTA-2. Given that our attempts were unsuccessful, we employed a static docking approach using the published AtDHDPR2 crystal structure (*Watkin et al., 2018*). The resulting data suggested that MBDTA-2 binds in the active site with a binding affinity of −6.2 kcal mol$^{-1}$ (*Figure 4*). The hydrophobic pocket occupied by MBDTA-2 overlaps with the probable NADPH cofactor-binding site, based on the crystal structure of cofactor-bound *Escherichia coli* DHDPR (*Reddy et al., 1996*; *Scapin et al., 1997*). The predicted MBDTA-2 orientation suggests its stabilisation by polar interactions between the heterocyclic ring and Thr122 and Gly120. Additionally, the MBDTA-2 acid is within hydrogen bonding proximity to Asp185. To validate the mechanism of inhibition of MBDTA-2 against AtDHDPR, further enzyme kinetic experiments were performed. The previous dose–response experiments were conducted according to standard practice in that substrate and cofactor were kept at limiting concentrations to ensure that inhibition may be measured regardless of the kinetic mechanism of inhibition. Subsequently, the activity of AtDHDPR was measured whilst titrating MBDTA-2, this time in the presence of excess amounts of substrate and nucleotide cofactor, that is at concentrations 10-fold above the respective $K_M$ values (*Figure 4B*). The IC$_{50}$ values determined were 72.7 ± 1.07 and 69.5 ± 1.06 µM for AtDHDPR1 and AtDHDPR2, respectively, which are 10- and 8-fold greater than those determined for AtDHDPR1 and AtDHDPR2 when substrate and cofactor were limiting. This apparent reduction in potency indicates that MBDTA-2 is a competitive inhibitor, and therefore likely binds at the AtDHDPR active site as suggested by the docking results. Interestingly, this contrasts with the allosteric mode of inhibition of MBDTA-2 against AtDHDPS.

## Herbicidal activity of MBDTA-2 against weeds

Previously, we showed that the MBDTA-2 compound has herbicidal activity against the model plant *A. thaliana* and is therefore the first example of a herbicidal lysine biosynthesis inhibitor. To further

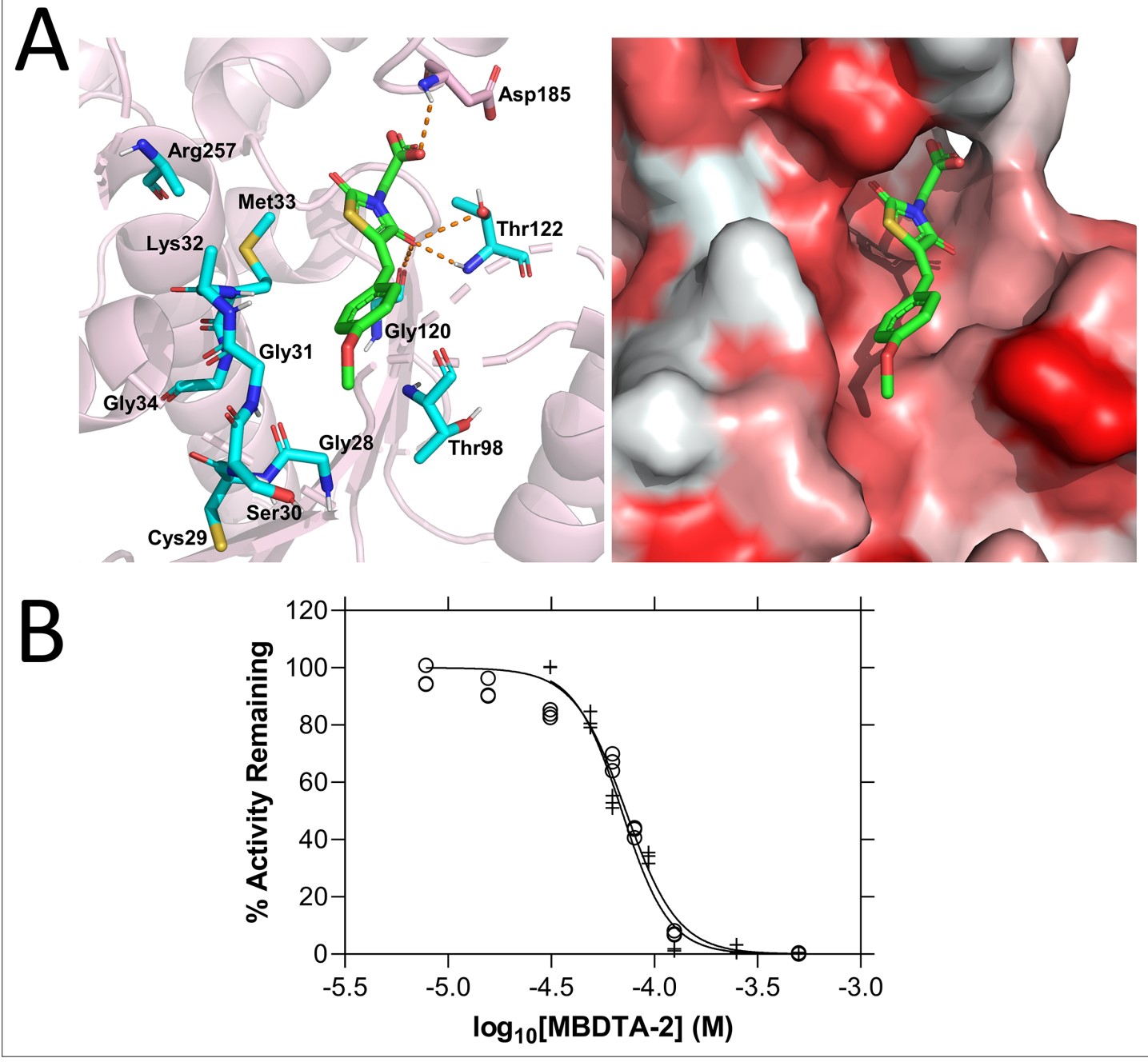

**Figure 4.** Mode of AtDHDPR2 inhibition by (*Z*)-2-(5-(4-methoxybenzylidene)-2,4-dioxothiazolidin-3-yl)acetic acid (MBDTA-2). (**A**) The predicted MBDTA-2 (green)-binding site resulting from static docking with AtDHDPR2 (PDB ID: 5UA0) overlaps with the probable NADPH cofactor-binding site (cyan, left panel). Hydrophobicity of the predicted binding pocket (right panel) is represented by white-red shading indicating hydrophilic–hydrophobic residues. (**B**) Dose–response curves of MBDTA-2 against AtDHDPR1 (○) and AtDHDPR2 (+) enzymes in the presence of saturating concentrations of substrate and cofactor. Data were fitted to a nonlinear regression model (solid line), resulting in $R^2$ values of 0.97 and 0.98 for AtDHDPR1 and AtDHDPR2, respectively.

The online version of this article includes the following source data for figure 4:

**Source data 1.** Source data for *Figure 4*.

assess the potential of inhibiting plant lysine biosynthesis enzymes for the development of herbicides, the efficacy of MBDTA-2 against the economically significant invasive weed species rigid ryegrass *L. rigidum* was investigated. Treatment of *L. rigidum* with 1200 mg l⁻¹ of MBDTA-2 resulted in inhibition of plant germination and growth, corresponding to a significant reduction in shoot fresh and dry weight and a significant reduction in root dry weight (*Figure 5*). Specifically, we observed ~4- and

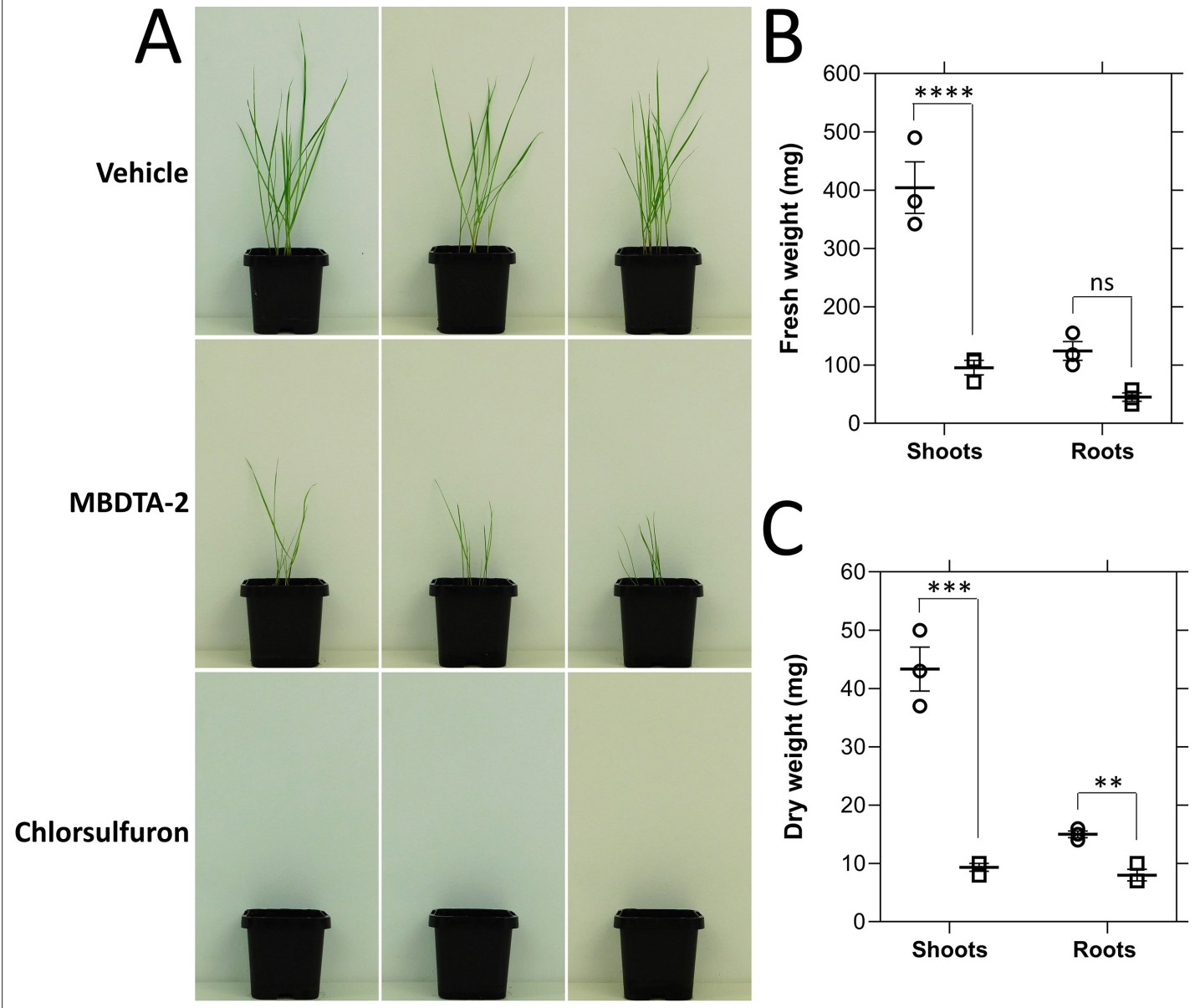

**Figure 5.** Inhibition of *Lolium rigidum* germination and growth by (*Z*)-2-(5-(4-methoxybenzylidene)-2,4-dioxothiazolidin-3-yl)acetic acid (MBDTA-2). (**A**) Day growth of *L. rigidum* treated with three pre-emergence treatments of vehicle control (2% (vol/vol) DMSO, 0.01% Agral), or 1200 mg $l^{-1}$ of MBDTA-2, or 1200 mg $l^{-1}$ of chlorsulfuron. Treatments were given by pipetting 2.0 ml per pot directly onto seeds. (**B**) Fresh weight of *L. rigidum* shoots and roots following treatment of plants with vehicle control (dots) or MBDTA-2 (lines). Shoots, p = 0.00002, roots, p = 0.05233, unpaired Student's two-tailed *t*-test. (**C**) Dry weight of *L. rigidum* shoots and roots following treatment of plants with vehicle control (dots) or MBDTA-2 (lines). Shoots, p = 0.00088, roots, p = 0.00374, unpaired Student's two-tailed *t*-test. Data were normalised against the vehicle control. Data represent mean ± standard error of the mean (SEM) (*N* = 3). **p < 0.01, ***p < 0.001, ****p < 0.0001.

The online version of this article includes the following source data for figure 5:

**Source data 1.** Source data for *Figure 5*.

~5-fold reductions in shoot fresh and dry weight, respectively, and a ~2-fold reduction in root dry weight (*Figure 5*). These results further exemplify the potential of lysine biosynthesis inhibitors for development as herbicide candidates.

## Discussion

Herbicides with new modes of action are urgently needed to combat the rise in herbicide-resistant weed species, which pose a global threat to agriculture. In our previous study, we described the development of the first herbicidal lysine biosynthesis inhibitors, providing proof-of-concept for a novel herbicide mode of action. Whilst we previously explored the molecular mode of action of these inhibitors at their target site, namely the DHDPS enzyme, an explanation for their unusual similarity in *in vitro* and *in vivo* potency remained to be delineated. Specifically, we hypothesised that they may be acting as proherbicides, through demethylation of their aryl methyl ethers. While this mechanism of proherbicide conversion has not been reported to date, the demethylation of aryl methyl ethers on prodrugs, such as codeine, is well established (*Dayer et al., 1988*; *Kirchheiner et al., 2007*). Nevertheless, our finding that the demethylated analogue of the MBDTA-2 aryl methyl ether did not positively impact activity against the target enzyme DHDPS demonstrated that these compounds are not proherbicides.

Given that we could not attribute the similarity between the *in vitro* and *in vivo* potency of our compounds to their modification to a more active form *in vivo*, we postulated that we may have previously failed to capture the totality of their target site effects. Our discovery that MBDTA-2 is an inhibitor of not only DHDPS, but also of the subsequent enzyme in the lysine biosynthesis pathway DHDPR, supported this hypothesis. Moreover, the ~8-fold greater potency of MBDTA-2 against DHDPR than DHDPS reveals that the *in vitro* potency is actually ~6-fold greater than the *in vivo* potency. Furthermore, the low-micromolar potency of MBDTA-2 at the DHDPR target site is comparable to the potency of glyphosate, the most successful commercial herbicide active ingredient, at its enzyme target 5-enolpyruvylshikimate-3-phosphate synthase (*Mao et al., 2016*; *Sammons et al., 2018*). The phenomenon of inhibitors having dual-target activity against consecutive enzymes in metabolic pathways has sometimes been attributed to conserved active site features (*Hsu et al., 2013*; *Toulouse et al., 2020*). However, there are also many examples of inhibitors of multiple targets from distinct pathways, which have been identified regardless of binding site similarities (*Allen et al., 2015*; *Hashmi et al., 2021*; *Wang et al., 2019*). Nevertheless, to our knowledge, this is the first time a dual-target inhibitor has been shown to have allosteric and orthosteric inhibitory activity at two different enzymes. The advantages of multi-target inhibitors over single-target inhibitors have been well reported for the development of novel drugs and fungicides (*Allen et al., 2015*; *Gullino et al., 2010*; *Oldfield and Feng, 2014*). Such advantages include a reduced susceptibility to the generation of resistance, which is also a highly desirable property in herbicide development (*Gressel, 2020*; *Hall et al., 2020*). Indeed, there has been a focus on the use of herbicide mixtures inhibiting multiple molecular targets in attempts to reduce the generation of resistance to existing herbicides (*Gressel, 2020*). Despite the recognition of the potential of inhibiting multiple targets for the reduction of target site resistance generation, little work has been done on the development of new herbicides that do so (*Fu et al., 2019*; *Giberti et al., 2017*). Dual-target compounds such as MBDTA-2 are therefore promising candidates for progressing the herbicide development field beyond the 'one target-one herbicide' approach.

For the future development of dual-target herbicides, the previously published DHDPS co-crystal structure and the DHDPR-binding model presented here could be used for the rational design of new MBDTA-2 analogues with increased target site activity (*Soares da Costa et al., 2021*). Such rational design efforts could also be guided by additional kinetic assays in the presence of sub-saturating amounts of MBDTA-2. Examining any changes in the $K_M$ values for the DHDPR substrate and cofactor under these conditions may provide further insights into MDBTA-2 interactions at the binding site. Additionally, exploring the structure–activity relationship of MBDTA-2 through the screening of analogues against whole plants may also provide insights into modifications that are advantageous to *in vivo* potency.

Although these methods offer opportunities to optimise inhibitor potency, formulation also has a substantial impact on the performance of herbicide active ingredients and is therefore an important avenue to pursue for potency maximisation. Whilst the *in vivo* assays conducted here provide proof-of-concept for the potential of dual-target lysine biosynthesis inhibitors as herbicide active ingredients,

controlled-dosage spraying experiments with the formulated compound will be pertinent to assess the application rate required for herbicidal efficacy in comparison to commercial herbicides. Furthermore, metabolomics experiments to quantify changes in lysine and other aspartate-derived amino acid levels in response to treatment with dual-target lysine biosynthesis inhibitors would be of interest to validate the mode of action at the target site, as well as elucidate whether pathway deregulation contributes to the *in vivo* activity.

The development of new herbicides that are effective against *L. rigidum*, particularly those with a reduced propensity to generate resistance, is of the highest priority given the economic impact of this species. An overreliance on a small number of herbicide modes of action has culminated in the widespread evolution of multiple resistance mechanisms in *L. rigidum* (**Bajwa et al., 2021**; **Owen et al., 2014**). In Australia alone, herbicide-resistant *L. rigidum* invades 8 million hectares of cropping land, resulting in revenue losses of AUD$93 million annually (**Llewellyn et al., 2016**). Our finding that MBDTA-2 can significantly reduce *L. rigidum* germination and growth further illustrates the potential utility of lysine biosynthesis inhibitors in combatting the global herbicide resistance crisis.

## Materials and methods

**Key resources table**

| Reagent type (species) or resource | Designation | Source or reference | Identifiers | Additional information |
|---|---|---|---|---|
| Gene (*Arabidopsis thaliana*) | DHDPS1 | TAIR | AtG60880 | |
| Gene (*Arabidopsis thaliana*) | DHDPS2 | TAIR | AtG45440 | |
| Gene (*Arabidopsis thaliana*) | DHDPR1 | TAIR | At2G44040 | |
| Gene (*Arabidopsis thaliana*) | DHDPR2 | TAIR | At3G59890 | |
| Software, algorithm | PyRX | Source Forge | | Version 0.8 |

### Chemical synthesis

Compounds were synthesised as previously described (**Christoff et al., 2021**; **Perugini et al., 2018**).

### Protein expression and purification

Recombinant AtDHDPS1, AtDHDPS2, AtDHDPR1, and AtDHDPR2 proteins were produced as previously described (**Mackie et al., 2022**; **Soares da Costa et al., 2021**).

### Enzyme inhibition assays

DHDPS enzyme activity was measured using methods previously described (**Soares da Costa et al., 2021**). DHDPR enzyme activity was measured using methods previously described (**Mackie et al., 2022**). Briefly, reaction mixtures were incubated at 30°C for 12 min before a second 60-s incubation following the addition of excess *E. coli* DHDPS (51 µg ml$^{-1}$) for generation of the DHDP substrate. The relevant DHDPR isoform (2.6 µg ml$^{-1}$) was added to initiate the reaction, and substrate turnover measured spectrophotometrically at 340 nm via the associated oxidation of the cofactor NADPH. Experiments were performed in technical triplicates.

### Docking

The AtDHDPR2 crystal structure was retrieved from the Protein Data Bank and hydrogens added using AutoDock Tools. Three-dimensional MBDTA-2 was docked with AtDHDPR2 using an unlimited search space in the PyRX interface using AutoDock Vina with default parameters. The resulting ligand poses were visualised in PyMol.

### Herbicidal activity analyses

The herbicidal efficacy of MBDTA-2 against *L. rigidum* was assessed using methods similar to those reported previously (**Mackie et al., 2022**). Pre-wet seed-raising soil (pH 5.5) (Biogro, Dandenong South, VIC, Australia) supplemented with 0.22% (wt/wt) Nutricote N12 Micro 140 day-controlled release fertiliser (Yates, Sydney, NSW, Australia) was used. Ten seeds were sown at a depth of 0.5 cm into pots of pre-wet soil, following stratification at 4°C for 21 days in the dark. Compounds dissolved

in DMSO were diluted to working concentrations in $H_2O$ containing 0.01% (vol/vol) Agral (Syngenta, North Ryde, NSW, Australia) to a final DMSO concentration of 2% (vol/vol). Treatments were given by pipetting 2.0 ml of MBDTA-2, vehicle control or positive control (chlorosulfuron PESTANAL [Sigma-Aldrich, North Ryde, NSW, Australia]) directly onto seeds upon sowing and on each of the subsequent 2 days. Plants were grown in a chamber at 22°C under a 16 hr light (100 µmol $m^{-2}$ $s^{-1}$)/8 hr dark schedule for 14 days before photos were taken. Roots and shoots were separated prior to drying at 70°C for 72 hr. Experiments were performed in biological triplicates.

## Acknowledgements

T.P.S.C. acknowledges the Australian Research Council for funding support through a DECRA (DE190100806) and The University of Adelaide for a Future Making Fellowship. Work in A.R.G.'s laboratory is supported by the Australian Research Council Research Hub for Medicinal Agriculture (IH180100006). E.R.R.M. acknowledges the Grains Research and Development Corporation (9176977) for support through a PhD scholarship and operational funding. E.R.R.M. and R.M.C. are recipients of Australian Government Research Training Program Scholarships. R.M.C. is also the recipient of a LIMS Write-Up Award.

## Additional information

### Competing interests

Belinda M Abbott: B.M.A. are listed as inventors on a patent pertaining to inhibitors described in the manuscript. Patent Title: Heterocyclic inhibitors of lysine biosynthesis via the diaminopimelate pathway; International patent (PCT) No.: WO2018187845A1; Granted: 18/10/2018. Tatiana P Soares da Costa: T.P.S.C. is listed as inventors on a patent pertaining to inhibitors described in the manuscript. Patent Title: Heterocyclic inhibitors of lysine biosynthesis via the diaminopimelate pathway; International patent (PCT) No.: WO2018187845A1; Granted: 18/10/2018. The other authors declare that no competing interests exist.

### Funding

| Funder | Grant reference number | Author |
| --- | --- | --- |
| Australian Research Council | DE190100806 | Tatiana P Soares da Costa |
| Australian Research Council | IH180100006 | Anthony R Gendall |
| Grains Research and Development Corporation | 9176977 | Emily RR Mackie |

The funders had no role in study design, data collection, and interpretation, or the decision to submit the work for publication.

### Author contributions

Emily RR Mackie, Data curation, Formal analysis, Investigation, Methodology, Validation, Visualization, Writing – original draft, Writing – review and editing; Andrew S Barrow, Rebecca M Christoff, Data curation, Formal analysis, Investigation, Methodology, Validation, Writing – review and editing; Belinda M Abbott, Anthony R Gendall, Methodology, Resources, Supervision, Writing – review and editing; Tatiana P Soares da Costa, Conceptualization, Funding acquisition, Methodology, Project administration, Resources, Supervision, Writing – original draft, Writing – review and editing

### Author ORCIDs

Emily RR Mackie http://orcid.org/0000-0002-0851-5741
Andrew S Barrow http://orcid.org/0000-0003-3661-5173
Rebecca M Christoff http://orcid.org/0000-0003-2111-3218
Anthony R Gendall http://orcid.org/0000-0002-2255-3939
Tatiana P Soares da Costa http://orcid.org/0000-0002-6275-7485

Decision letter and Author response
Decision letter https://doi.org/10.7554/eLife.78235.sa1
Author response https://doi.org/10.7554/eLife.78235.sa2

## Additional files

### Supplementary files
• Transparent reporting form

### Data availability
Figure 2 - Source Data 1, Figure 3 - Source Data 1, Figure 4 - Source Data 1 and Figure 5 - Source Data 1 contain the numerical data used to generate the figures.

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
