## [Editor Report]

This paper presents the highly interesting and novel finding that the investigated compound inhibits two targets in lysine synthesis. Further, the first enzyme has allosteric inhibition, while the second enzyme is a competitive inhibitor. The authors do a good job explaining why this is high interest for herbicide resistance management in a new compound. The authors demonstrated that the compound is not a pro-herbicide, and propose that the higher in vivo activity relative to in vitro activity is due to the simultaneous in vivo inhibition of two separate steps in lysine synthesis.

---

## [Decision Letter]

**Decision letter after peer review:**

Thank you for submitting your article "A dual-target herbicidal inhibitor of lysine biosynthesis" for consideration by *eLife*. Your article has been reviewed by 3 peer reviewers, including Todd Gaines as Reviewing Editor and Reviewer #1, and the evaluation has been overseen by Detlef Weigel as the Senior Editor. The following individual involved in review of your submission has agreed to reveal their identity: Paul Schmitzer (Reviewer #2).

Essential revisions:

1) Please provide the chlorsulfuron positive control data for the Lolium rigidum bioassay figure. Alter the claims for biological activity on Lolium rigidum and remove the extrapolated application rate, or conduct a new bioassay with methods suggested in the reviews.

2) The question arose in the reviews as to whether the compound is a pro-herbicide with respect to DHPDR. Please alter the discussion to raise this question, or consider conducting the experiment to add the demethylated compound to the DHPDR inhibition assay to test its activity.

*Reviewer #1 (Recommendations for the authors):*

Excellent paper. The authors could include a reference to work in fungicides to provide further evidence that multi-site inhibitors are lower risk for resistance evolution than single-site inhibitors. The bioassay method used for Lolium rigidum is unusual. The authors pipetted 2 mL of a 1200 mg/L solution directly onto seeds when sowing. Alternatives that could be used would include soaking seeds for 24 hour in herbicide solution; using an agarose based assay; spraying post-emergence on emerged seedlings; or spraying directly onto seeds or seedlings in soil. The authors report a calculated application equivalent of 48 kg/ha. It is not clear how this extrapolated number is obtained, nor is it realistic. No pesticide could be applied at such a high rate (the compound would be coated like salt onto everything in the field), and the sheer volume of water application required to achieve that would also represent drenching a field. I suggest that the authors remove this extrapolated calculation, and perhaps instead discuss the relative activity/toxicity of their compound in comparison to other known herbicides based on in vitro inhibition of the two enzymes. I don't think that more experiments are needed for this paper, but future work could conduct bioassays using methods that would provide more accurate ways to extrapolate to potential field use rates. Future work could also conduct biochemical measurements to ask whether lysine is reduced in Lolium rigidum following treatment with this herbicide, supporting the prediction that the mode of action is two-site inhibition of lysine synthesis. I also suggest that the authors provide more methods in the figure caption for Figure 5 so that it stand alone. Currently the caption only states the application concentration of 1200 mg/L, but it does not provide the application volume of 2 mL pipetted onto each seed, so in the current presentation the reader must examine the methods to understand the total dose applied to the plants. Suggest adding the 2 mL pipetting onto each seed to the caption.

*Reviewer #2 (Recommendations for the authors):*

The paper was a joy to read as it represents many processes in the design of new herbicides. I don't like creating additional work but through my review of the paper I kept wondering if the analogs presented in your previous publications (FBDTA and MBDTA-1), as well as any other available analogs, could build the knowledge of the inhibition of DHDPR and show a limited SAR. This may be a topic for a future publication so please disregard it if not appropriate for this paper.

*Reviewer #3 (Recommendations for the authors):*

This is a well-written description of experiments to further determine the mode of action of novel lysine synthesis inhibitor. The expansion of the mode of action to two consecutive enzymes of the lysine synthesis pathway is a surprising result, especially with different binding modes for the two enzymes. Again, the in vitro activity MBDTA-2 is rather poor for an effective herbicide. Perhaps inhibition of two enzymes makes up for the poor activity on each of the enzymes. Was the demethylated form of the compound tested in an in vitro assay of AtDHDPR1 and AtDHDP2? This must be done. If HBDTA is much more active than MBDTA-2 on DHDPR, the results would make more sense to me.

The bioassay method is very weak, as it is impossible to determine whether MBDTA-2 is an effective herbicide from these results. Herbicides are not pipetted onto weed seeds in the field. MBDTA-2 was applied three times, on three consecutive days. In the field, a herbicide is applied once. I suggest that the compounds be applied once to the soil containing weeds seeds and watered into soil, as soil applied herbicide are used, or applied to foliage as a foliar spray with an adjuvant. I would try both methods, as some herbicides that are excellent soil-applied herbicides are poor foliar herbicides and vice versa. This way, you can compare the efficacy in g/ha with the positive control. Finally, herbicides are expected to kill weeds. Figure 5 shows only growth stunting. Where are the chlorsulfuron results?

---

## [Author Response]

Essential revisions:1) Please provide the chlorsulfuron positive control data for the Lolium rigidum bioassay figure. Alter the claims for biological activity on Lolium rigidum and remove the extrapolated application rate, or conduct a new bioassay with methods suggested in the reviews.

The chlorsulfuron positive control data has been added to Figure 5, panel A. The Figure 5 legend has been updated accordingly with the addition of ‘or 1200 mg·L^-1^ of chlorsulfuron.’ The claims for biological activity on *Lolium rigidum* have been altered as follows: Line 216: ‘as we have demonstrated that MBDTA-2 possesses herbicidal activity against one of the most problematic weed species to global agriculture’ deleted; Line 81:

“we successfully extended our previous herbicidal activity studies’ changed to ‘we extended our previous in vivo activity studies’; Line 271: ‘herbicidal’ deleted. Line 211: the extrapolated application rate has been removed by deleting ‘(equivalent to 48 kg·ha^-1^)”.

2) The question arose in the reviews as to whether the compound is a pro-herbicide with respect to DHPDR. Please alter the discussion to raise this question, or consider conducting the experiment to add the demethylated compound to the DHPDR inhibition assay to test its activity.

The inhibition of DHDPR by the demethylated compound has now been measured and added to Figure 3B and described in the Results section starting at Line 134: Given that these results revealed a new target site for MBDTA-2, we assessed whether the compound could be proherbicidal with respect to DHDPR. To do so, we assessed the inhibitory activity of HBDTA against both AtDHDPR isoforms to determine the IC_50_ values of 169 ± 0.92 µM and 155.3 ± 0.90 µM for AtDHDPR1 and AtDHDPR2, respectively (Figure 3B). HBDTA is therefore >18-fold less potent at the DHDPR target site relative to MBDTA-2, suggesting that the MBDTA-2 methyl group is beneficial for DHDPR inhibition. As such, MBDTA-2 is unlikely to be a proherbicide.

Reviewer #1 (Recommendations for the authors):Excellent paper. The authors could include a reference to work in fungicides to provide further evidence that multi-site inhibitors are lower risk for resistance evolution than single-site inhibitors.

We thank the reviewer for the feedback. Line 263: ‘and fungicides’ and Gullino et al., 2010 reference added to provide further evidence that multi-site inhibitors are lower risk for resistance evolution than single-site inhibitors. Line 262: ‘dual-target’ changed to ‘multi-target’ to encompass the use of multi-site fungicides.

The bioassay method used for Lolium rigidum is unusual. The authors pipetted 2 mL of a 1200 mg/L solution directly onto seeds when sowing. Alternatives that could be used would include soaking seeds for 24 hour in herbicide solution; using an agarose based assay; spraying post-emergence on emerged seedlings; or spraying directly onto seeds or seedlings in soil.

We thank the reviewer for the insightful comment. Direct pipetting onto seeds on soil, as employed in this study, is similar to soaking the seeds prior to planting, but is more relevant to herbicide use in the field. Pipetting directly onto seeds also functions similarly to spraying directly onto the seeds, however it provides a more consistent and controlled dosage as over-spray can occur with spray application. Experiments with Arabidopsis indicated that these inhibitors are more effective when applied pregermination. The value of further bioassays, as highlighted by the reviewer, has been acknowledged with the addition of the following at Line 287:

“Whilst the in vivo assays conducted here provide proof-of concept for the potential of dual-target lysine biosynthesis inhibitors as herbicide active ingredients, controlled-dosage spraying experiments with the formulated compound will be pertinent to assess the application rate required for herbicidal efficacy in comparison to commercial herbicides.”

The authors report a calculated application equivalent of 48 kg/ha. It is not clear how this extrapolated number is obtained, nor is it realistic. No pesticide could be applied at such a high rate (the compound would be coated like salt onto everything in the field), and the sheer volume of water application required to achieve that would also represent drenching a field. I suggest that the authors remove this extrapolated calculation, and perhaps instead discuss the relative activity/toxicity of their compound in comparison to other known herbicides based on in vitro inhibition of the two enzymes.

We thank the reviewer for the comment. Line 211: The extrapolated application rate has been removed by deleting ‘(equivalent to 48 kg·ha^-1^)’. The relative in vitro activity of our compound in comparison to known herbicides is now discussed with the following addition at Line 252:

“Furthermore, the low-μM potency of MBDTA-2 at the DHDPR target site is comparable to the potency of glyphosate, a commercial herbicide active ingredient, at its enzyme target 5-enolpyruvylshikimate-3phosphate synthase (Mao et al., 2016; Sammons et al., 2018)”.

I don't think that more experiments are needed for this paper, but future work could conduct bioassays using methods that would provide more accurate ways to extrapolate to potential field use rates. Future work could also conduct biochemical measurements to ask whether lysine is reduced in Lolium rigidum following treatment with this herbicide, supporting the prediction that the mode of action is two-site inhibition of lysine synthesis.

We thank the reviewer for their feedback. The raised points about future work have now been added to the Discussion as follows: Line 287: the value of further bioassays has been acknowledged with the addition of ‘Whilst the in vivo assays conducted here provide proof-of-concept for the potential of dual-target lysine biosynthesis inhibitors as herbicide active ingredients, controlled-dosage spraying experiments with the formulated compound will be pertinent to assess the application rate required for herbicidal efficacy in comparison to commercial herbicides.’ Line 291:

“quantification of lysine levels following treatment with these compounds has been added as a future direction by adding ‘Furthermore, metabolomics experiments to quantify changes in lysine and other aspartate-derived amino acid levels in response to treatment with dual-target lysine biosynthesis inhibitors would be of interest to validate the mode of action at the target site, as well as elucidate whether pathway deregulation contributes to the in vivo activity.”

I also suggest that the authors provide more methods in the figure caption for Figure 5 so that it stand alone. Currently the caption only states the application concentration of 1200 mg/L, but it does not provide the application volume of 2 mL pipetted onto each seed, so in the current presentation the reader must examine the methods to understand the total dose applied to the plants. Suggest adding the 2 mL pipetting onto each seed to the caption.

We thank the reviewer for the comment. Line 221: ‘Treatments were given by pipetting 2.0 mL per pot directly onto seeds’ added to Figure 5 legend.

Reviewer #2 (Recommendations for the authors):The paper was a joy to read as it represents many processes in the design of new herbicides. I don't like creating additional work but through my review of the paper I kept wondering if the analogs presented in your previous publications (FBDTA and MBDTA-1), as well as any other available analogs, could build the knowledge of the inhibition of DHDPR and show a limited SAR. This may be a topic for a future publication so please disregard it if not appropriate for this paper.

Indeed, future work will involve the synthesis of a comprehensive series of analogues to understand the structure-activity relationship of dual-target DHDPS/DHDPR inhibitors. This has been added to the Discussion at Line 281 as follows:

“Additionally, exploring the structure-activity relationship of MBDTA-2 through the screening of analogues against whole plants may also provide insights into modifications that are advantageous to in vivo potency.”

Reviewer #3 (Recommendations for the authors):This is a well-written description of experiments to further determine the mode of action of novel lysine synthesis inhibitor. The expansion of the mode of action to two consecutive enzymes of the lysine synthesis pathway is a surprising result, especially with different binding modes for the two enzymes. Again, the in vitro activity MBDTA-2 is rather poor for an effective herbicide. Perhaps inhibition of two enzymes makes up for the poor activity on each of the enzymes. Was the demethylated form of the compound tested in an in vitro assay of AtDHDPR1 and AtDHDP2? This must be done. If HBDTA is much more active than MBDTA-2 on DHDPR, the results would make more sense to me.

We thank the reviewer for raising an interesting point. The IC_50_ values of MBDTA-2 against AtDHDPR1 and AtDHDPR2 are 6.9 and 8.6 µM, respectively. These values are similar to the IC_50_ of the herbicide active ingredient glyphosate against its enzyme target. The potency of MBDTA-2 relative to current herbicides is now discussed with the addition of Line 252: ‘Furthermore, the low-μM potency of MBDTA-2 at the DHDPR target site is comparable to the potency of glyphosate, the most successful commercial herbicide active ingredient, at its enzyme target 5-enolpyruvylshikimate-3-phosphate synthase.’ We have now tested the demethylated compound against AtDHDPR1 and AtDHDPR2 and included the results in the revised manuscript as described in Response 2 above.

The bioassay method is very weak, as it is impossible to determine whether MBDTA-2 is an effective herbicide from these results. Herbicides are not pipetted onto weed seeds in the field. MBDTA-2 was applied three times, on three consecutive days. In the field, a herbicide is applied once. I suggest that the compounds be applied once to the soil containing weeds seeds and watered into soil, as soil applied herbicide are used, or applied to foliage as a foliar spray with an adjuvant. I would try both methods, as some herbicides that are excellent soil-applied herbicides are poor foliar herbicides and vice versa. This way, you can compare the efficacy in g/ha with the positive control. Finally, herbicides are expected to kill weeds. Figure 5 shows only growth stunting.

The bioassays conducted in this study provide proof-of-concept work for the potential utility of dual-target lysine biosynthesis inhibitors for development as herbicides. Future work will focus on optimisation of the inhibitors presented in this study in order to develop them as herbicides. This point is now included in the Discussion at line 281 as follows: ‘Additionally, exploring the structure activity relationship of MBDTA-2 through the screening of analogues against whole plants may also provide insights into modifications that are advantageous to in vivo potency.’ Future work will also optimise the formulation of these compounds to improve efficacy and this is included in the Discussion at Line 285 as follows: ‘Although these methods offer opportunities to optimise inhibitor potency, formulation also has a substantial impact on the performance of herbicide active ingredients and is therefore an important avenue to pursue for potency maximisation (Pacanoski, 2015). Whilst the in vivo assays conducted here provide proof-of-concept for the potential of dual-target lysine biosynthesis inhibitors as herbicide active ingredients, controlled-dosage spraying experiments with the formulated compound will be pertinent to assess the application rate required for herbicidal efficacy in comparison to commercial herbicides.’ The compounds presented demonstrate that they inhibit the germination and growth of a weed species and therefore could potentially be developed into herbicides with future optimisation. Accordingly, the claims for biological activity on *Lolium rigidum* have been altered as discussed in Response 1 above.

Where are the chlorsulfuron results?

The chlorsulfuron positive control data has been added to Figure 5, panel A. Line 220:

“The Figure 5 legend has been updated accordingly with the addition of ‘or 1200 mg·L^-1^ of chlorsulfuron.”